# Development and Validation of the HIV-CARDIO-PREDICT Score to Estimate the Risk of Cardiovascular Events in HIV-Infected Patients

**DOI:** 10.3390/cells12040523

**Published:** 2023-02-05

**Authors:** Styliani Karanika, Theodoros Karantanos, Herman Carneiro, Sabrina A. Assoumou

**Affiliations:** 1Internal Medicine Department, Boston Medical Center, Boston, MA 02118, USA; 2School of Medicine, Division of Infectious Diseases, The Johns Hopkins Hospital, Baltimore, MD 21287, USA; 3Department of Medical Oncology, Hematologic Malignancies, Sidney Kimmel Cancer Center, The Johns Hopkins Hospital, Baltimore, MD 21287, USA; 4Department of Medicine, Division of Cardiology, Case Western Reserve University/University Hospitals Cleveland Medical Center, Cleveland, OH 44106, USA; 5Department of Medicine, Section of Infectious Diseases, Boston University School of Medicine, Boston, MA 02118, USA; 6Department of Medicine, Section of Infectious Diseases, Boston Medical Center, Boston, MA 02118, USA

**Keywords:** HIV, cardiovascular disease risk, prediction tool, risk score

## Abstract

Importance: Commonly used risk assessment tools for cardiovascular disease might not be accurate for HIV-infected patients. Objective: We aimed to develop a model to accurately predict the 10-year cardiovascular disease (CV) risk of HIV-infected patients. Design: In this retrospective cohort study, adult HIV-infected patients seen at Boston Medical Center between March 2012 and January 2017 were divided into model development and validation cohorts. Setting: Boston Medical Center, a tertiary, academic medical center. Participants: Adult HIV-infected patients, seen in inpatient and outpatient setting. Main Outcomes and Measures: We used logistic regression to create a prediction risk model for cardiovascular events using data from the development cohort. Using a point-based risk-scoring system, we summarized the relationship between risk factors and cardiovascular disease (CVD) risk. We then used the area under the receiver operating characteristics curve (AUC) to evaluate model discrimination. Finally, we tested the model using a validation cohort. Results: 1914 individuals met the inclusion criteria. The model had excellent discrimination for CVD risk [AUC 0.989; (95% CI: 0.986–0.993)] and included the following 11 variables: male sex (95% CI: 2.53–3.99), African American race/ethnicity (95% CI: 1.50–3.13), current age (95% CI: 0.07–0.13), age at HIV diagnosis (95% CI: −0.10–(−0.02)), peak HIV viral load (95% CI: 9.89 × 10^−7^–3.00 × 10^−6^), nadir CD4 lymphocyte count (95% CI: −0.03–(−0.02)), hypertension (95% CI: 0.20–1.54), hyperlipidemia (95% CI: 3.03–4.60), diabetes (95% CI: 0.61–1.89), chronic kidney disease (95% CI: 1.26–2.62), and smoking (95% CI: 0.12–2.39). The eleven-parameter multiple logistic regression model had excellent discrimination [AUC 0.957; (95% CI: 0.938–0.975)] when applied to the validation cohort. Conclusions and Relevance: Our novel HIV-CARDIO-PREDICT Score may provide a rapid and accurate evaluation of CV disease risk among HIV-infected patients and inform prevention measures.

## 1. Introduction

Antiretroviral therapy (ART) for human immunodeficiency virus (HIV) infection has improved the life expectancy of patients and decreased the prevalence of AIDS-related illnesses such as opportunistic infections [1]. Non-AIDS-related events such as cardiovascular disease and malignancies are increasingly becoming the primary cause of morbidity and mortality among HIV-infected patients [1,2,3]. Cardiovascular (CV) disease occurs earlier and at a higher frequency among HIV-infected patients compared with the general population [3,4,5,6,7]. It is unclear if this observation is primarily mediated by traditional risk factors such as hypertension (HTN), hyperlipidemia (HLD), or diabetes (DM) or if additional elements related to ART or chronic inflammation or immune activation are involved [8,9,10]. Studies have shown that commonly used cardiovascular prediction functions such as the Framingham Risk Score and American College of Cardiology/American Heart Association Atherosclerotic Cardiovascular Disease risk tool (ACC/AHA ASCVD) systematically underestimate risks among HIV-infected patients [11,12]. In addition, other risk prediction equations specifically developed for HIV-infected patients have been shown to underestimate cardiovascular risk when applied to different large cohorts of HIV-infected patients [13,14,15,16]. Furthermore, although commonly used indices such as the Veterans Aging Cohort Study Index (VACS Index) predicts all-cause mortality and incident myocardial infarction [17,18], we aimed to develop a tool that would predict additional cardiovascular events as defined by the American Heart Association (AHA). The latter incorporates myocardial infarctions and other events such as strokes, carotid endarterectomies, hospitalizations for unstable angina, and coronary bypass graft surgeries. Given calls by both infectious diseases and cardiology societies [11,19,20,21], we aimed to create a clinical tool to accurately predict cardiovascular risk among HIV-infected individuals by incorporating traditional CV disease risk factors and HIV-specific factors to guide clinical practice.

## 2. Methods

### 2.1. Study Design, Data Source, and Processing

We developed a retrospective cohort of laboratory-confirmed HIV-infected patients seen at Boston Medical Center (BMC) between 27 March 2012, and 31 January 2017. The start date was the publication date of the US Department of Health and Human Services guidance recommending ART for all HIV-infected patients. The entire cohort was divided into development (80%) and validation (20%) cohorts as per the Pareto principle (80–20 rule) [22]. The Boston Medical Center Institutional Review Board approved the study. A team of experienced physicians (SK, TK, and HC) abstracted the necessary information from two electronic medical record systems, given the overlap at the beginning of the study, to avoid missing any critical clinical data after their merge. We collected data on demographics, co-morbidities, HIV clinical characteristics, and cardiovascular events (CVEs). Inclusion criteria were as follows: (1) age ≥ 18 years at the start of the study; (2) laboratory-confirmed HIV diagnosis using antibody testing; (3) at least six months of follow-up time after the diagnosis of HIV. We excluded individuals with prior history of hypertension (HTN), hyperlipidemia (HLD), diabetes (DM), chronic kidney disease (CKD), and CVE at the time of the HIV diagnosis in an effort to not overestimate CV events given the retrospective design of our study. We also excluded “elite controllers”, separating this rare subset of HIV patients who are able to maintain an undetectable HIV viral load over time in the absence of antiretroviral therapy from the general HIV population. Last, patients with incomplete clinical information were kept out of our cohort. Details are shown in Figure 1.

Clinical diagnoses were obtained using International Statistical Classification of Diseases and Related Health Problems, 9th (ICD-9) or 10th version (ICD-10). They were supplemented by a detailed review of the medical chart (clinical notes of respective specialties including past medical, surgical, and social history, laboratory results, imaging studies, and procedure reports) by at least two reviewers. Co-morbidities (HTN, DM, HLD, and CKD) based on definitions provided by guidance panels or medical societies [20,23,24,25,26].

We defined the “time to reach an undetectable status” as the estimated number of months between ART initiation and HIV viral load reported as <20 copies per ml in the medical record. HIV viral load measures were measured at a minimum every 6–8 weeks until they reached an undetectable level. ART regimens were obtained from prescriptions and refills in the medical record. Adherence to antiretroviral treatment was assessed independently by at least two reviewers by chart review using self-reported missed doses, pharmacy refills, and pill counts during pharmacy visits (consolidated in an ordinal range from worse to the best adherence).

### 2.2. Potential Predictors

We included the following demographic information and medical history as potential predictors: current age (defined as the age of the patient at the end of the study; 31 January 2017), age at HIV diagnosis, gender, race/ethnicity, hypertension, hyperlipidemia, chronic kidney disease, diabetes, hepatitis C, tobacco use (defined as active smoking or cessation within the last five years since ASCVD Risk Estimator Plus considers former smokers’ CVD risk identical to never smokers after 5 years [27,28]), substance use disorder and alcohol use disorder. We also abstracted the following information on laboratory, clinical, or medication use: CD4 count at HIV diagnosis, CD4% at HIV diagnosis, nadir CD4 count, nadir CD4%, peak HIV viral load, HIV viral load at HIV diagnosis, number of months to reach an undetectable level, months from HIV diagnosis to a cardiovascular event, number of regimens required to achieve an undetectable level, adherence to ART, and treatment with abacavir for greater than six months. Potential predictors were selected based on existing literature on readily available demographics and clinical parameters associated with cardiovascular disease, HIV control, and HIV-associated inflammation.

### 2.3. Outcome

We used cardiovascular events as the outcome of interest. We defined outcomes using the AHA criteria: non-fatal myocardial infarctions including percutaneous coronary intervention, hospitalizations for unstable angina, coronary artery bypass graft surgeries (CABG), transient ischemic attacks, strokes, carotid endarterectomies, and sudden cardiovascular deaths [29].

### 2.4. Variable Selection and Score Construction

We used the development cohort for variable selection and risk score elaboration processes.

### 2.5. Statistical Analysis

First, we used the development cohort data to perform a bivariate logistic regression analysis evaluating the relationship between demographic and clinical characteristics and the outcome of interest, CVE. Using a stepwise analysis approach, we included statistically significant bivariate associations with potential clinical significance in a multivariable logistic regression model.

### 2.6. Assessment of Accuracy, Development of a Point System-Based Calculator, and Model Validation

We used the receiver operating characteristic analysis (ROC) to assess the model’s accuracy and logistic regression equations to develop a point-based risk-scoring system using a previously validated approach [30]. We organized statistically significant predictors into categories and determined each category’s referent risk factor profile. We then computed how far each risk factor category was from the base category in regression units. We also developed a point system to define the constant to reflect the increase in CV risk associated with a 10-year increase in age [30]. We determined the risk associated with each point and calculated the total points associated with each category following a validated approach [30]. We then developed score sheets to predict CVE for HIV-infected patients, and the total score ranged from 0 to 29.

Additionally, we tested the prediction rule derived from the development cohort data using the validation cohort. We also estimated the goodness of fit for all models with the Hosmer–Lemeshow test [31]. All *p*-values were two-sided, and we performed statistical analyses using STATA v13 (STATA Corp, College Station, TX, USA).

## 3. Results

### 3.1. Baseline Patient Characteristics

Of 3867 HIV-infected individuals seen at BMC during the time of interest, 1914 patients met the inclusion criteria. A total of 1531 (80%) HIV-infected patients were randomly selected for inclusion in the development cohort, and the rest, 383 (20%) patients, formed the validation cohort (Figure 1).

Table 1 shows the demographic and clinical characteristics of patients included in the development cohort. Six hundred and four (42.8%) were women. Most patients were African Americans (649, 42.4%), while individuals identifying as White were the second largest group (629, 41.1%). The mean (SD) age was 46.1 (11.2) years, and the mean (SD) age at HIV diagnosis was 37.1 (10.0) years. There were 384 documented cardiovascular events. The mean (SD) nadir CD4 lymphocyte count was 114.2 (57.4) cells/μL, and the mean (SD) peak HIV viral load was 280,149 (216,258) copies/mL. Appendix A features the demographic and clinical characteristics of the patients included in the development cohort.

### 3.2. Predictive Model

We performed bivariate logistic regression analyses, including potential predictors and CV events. The final predictive multivariable logistic regression model for CV risk was statistically significant (P_model_ < 0.001) and consisted of the following eleven parameters that were independently associated with this outcome: male sex (95% CI: 2.536–3.99, African American race/ethnicity (95% CI: 1.50–3.13), current age (95% CI: 0.07–0.13), age at HIV diagnosis (95% CI: −0.10–(−0.02)), peak HIV viral load (95% CI: 9.89 × 10^−7^–3.00 × 10^−6^), nadir CD4 lymphocyte count [95% CI: −0.03–(−0.02)], hypertension (95% CI: 0.20–1.54), hyperlipidemia (95% CI: 3.03–4.60), diabetes (95% CI: 0.61–1.89), chronic kidney disease (95% CI: 1.26–2.62), and smoking (95% CI: 0.12–2.39) (Table 2). ROC analysis identified this predictive model as an excellent discriminator of CVD risk [(AUC: 0.989, (95% CI: 0.986–0.993)] (Figure 2a). Sensitivity and specificity of the model were 75.10% and 94.27%, respectively, and positive and negative predictive values were 82.83% and 91.14%, respectively. In addition, the prediction rule demonstrated good statistical fitness (Hosmer–Lemeshow *p* = 1.00).

### 3.3. Estimating 10-Year Risk of Developing a CVE Using a Point System-Based Calculator

We estimated the ten-year risk of developing a CVE using a point system-based calculator, including statistically significant predictors. Table 3 shows the point system-based assigned to each risk factor. We then developed score sheets ranging from 0 to 29 to predict CVE for HIV-infected patients (HIV Cardiovascular Event risk score: HIV-CARDIO-PREDICT). Table 4 shows the estimated risk of developing cardiovascular events within 10 years per total points. For example, we calculated a score of 7 for a 40-year-old African American HIV-infected male without any other risk factors. This score corresponds to a 9.4% risk of developing a cardiovascular event within the next 10 years. The patient would be categorized as having intermediate risk (≥7.5 to ≤19.9) [19,32]. In comparison, the same patient would have had an ASCVD risk score of 2.3% [32], which corresponds to a low risk of developing a CVE in the subsequent 10 years (<5%), therefore potentially underestimating his actual risk.

### 3.4. Score Validation

The eleven-parameter multiple logistic regression model’s performance was also evaluated using the validation cohort. Baseline demographics and clinical characteristics for the validation cohort are available in the Appendix A. The model’s discrimination remained excellent [ROC AUC 0.957 (95% CI: 0.938–0.975)] (Figure 2b), and there was no evidence of poor fitness (Hosmer–Lemeshow *p* > 0.1). The model’s sensitivity and specificity were 78.45% and 94.10%, respectively, and positive and negative predictive values were 83.87% and 91.78%, respectively.

## 4. Discussion

We developed and validated a novel risk prediction tool (HIV-CARDIO-PREDICT Score) to estimate the ten-year cardiovascular risk of HIV-infected patients by including demographic, clinical, and laboratory information readily available in patients’ electronic health records. We found that older age, African American race/ethnicity, male gender, age at HIV diagnosis, lower nadir CD4 lymphocyte count, high peak HIV viral load, hypertension, hyperlipidemia, diabetes, chronic kidney disease, and tobacco use were associated with a higher risk of developing cardiovascular events as defined by the American Heart Association. The risk score had excellent discrimination for cardiovascular risk based on AUCs in development and validation cohorts of 0.99 and 0.96, respectively. This cardiovascular risk prediction model may help clinicians identify HIV-infected patients at higher risk and enable the early implementation of measures to prevent a significant cause of morbidity and mortality among HIV-infected patients who live longer due to antiviral therapy. Currently, clinicians use tools such as the ASCVD risk score to determine when to implement preventive measures such as aspirin or statins; however, these tools might miss patients who are eligible for these medications. For example, when using a hypothetical 40-year-old African American HIV-infected male without any other risk factors, we found that the management of this patient would change from categorizing him as low risk (<5% of developing a cardiovascular event within the next 10 years) using the ASCVD risk score to classifying him as intermediate risk (≥7.5 to ≤19.9%) according to our novel HIV-CARDIO-PREDICT Score.

In addition to finding an association between established risk factors such as hypertension, hyperlipidemia, diabetes, and smoking, we also uncovered additional clinical and laboratory findings that are key to predicting the future risk of cardiovascular events among HIV-infected patients. Our analysis revealed that the presence of chronic kidney disease, the nadir CD4 lymphocyte count, the peak viral load, and the age at HIV diagnosis were all crucial factors to include in the assessment. Notably, we determined that treatment with abacavir was not a significant factor in our model.

Prior studies in the general population have underscored the association between chronic kidney disease and cardiovascular disease [33]. In addition, a study using the Data Collection on Adverse Events of Anti-HIV Drugs (D:A:D) also showed a strong association between chronic kidney disease and cardiovascular disease among HIV-infected patients [34]. Furthermore, another study using the D:A:D cohort also revealed that lower CD4 lymphocyte count at baseline was an independent predictor of cardiovascular risk [13]. In an outpatient HIV clinical study, CD4 count <500 cells/mm^3^ was found to be is an independent risk factor for incident CVD [35] while in an independent HIV cohort endothelial dysfunction was associated with lower nadir CD4 count [36]. This literature supports our finding that nadir CD4 lymphocyte count is essential when determining cardiovascular risk among HIV-infected patients.

As for peak HIV viral load, prior studies have been mixed. The D:A:D study did not find a strong association between peak HIV viral load [14] and cardiovascular risk. Another study using data from a large US health care system showed that immunologic control was the most important HIV-related factor associated with acute myocardial infarction [37]. This study demonstrated that increased HIV viral load was not associated with acute myocardial infarction when CD4 was also considered in the model [37]. This contrasts our model, which determined that peak HIV viral load was independently associated with cardiovascular risk. Another randomized study showed that cardiovascular risk markers, including inflammatory, anti-inflammatory, and endothelial factors, were associated with HIV RNA replication [38]. This result provides some molecular data supporting our findings.

To the best of our knowledge, age at HIV diagnosis has not been extensively evaluated in prior studies focused on cardiovascular risk among HIV-infected patients. Our analysis showed that earlier HIV infection was an independent risk factor for the development of cardiovascular events. Our results are supported by findings suggesting that immune activation and inflammation are important drivers of cardiovascular disease. Evidence also shows that chronic inflammation persists even among virologically suppressed HIV-infected patients [39,40]. Studies, including HIV elite controllers, indicate that persistent inflammation may account for early cardiovascular disease [41,42].

Our finding that treatment with abacavir was not a strong predictor of cardiovascular events is in line with recent studies that did not show an association between abacavir use and myocardial infarction [43,44]. This contrasts with earlier analyses showing a worrisome association between abacavir treatment in observational studies [45]. This finding may be explained by the observation that fewer patients prefer abacavir nowadays when more modern regimens are available. Clinicians are also less likely to recommend abacavir for patients with a high pre-existing cardiovascular risk.

## 5. Limitations

There are some notable limitations to our study, including the use of a retrospective cohort from a single site with specific characteristics (an increased number of patients with substance use disorders is noted), which might limit generalizability. External validation of our findings using different cohorts of HIV-infected patients would provide information about the generalizability of our findings and support for clinical application. Moreover, accurate extraction of the clinical data can be challenging in a retrospective study; however, we tried to optimize the process by assigning at least two reviewers (medical providers) per clinical chart who assessed not only listed diagnoses through the ICD coding system but also considered data provided from clinical notes, laboratory results, and procedure reports. Patients with incomplete data were excluded as we were not able to accurately categorize their course. Another limitation is that we did not have access to death certificates and used information provided in the electronic medical records related to all outcomes considered. Last, we did not include integrase strand transfer inhibitors in the analysis as a possible risk factor of CVD because of the lack of concerning data until 2017. Newer information suggests that their use is associated with potential weight gain and development of cardiovascular disease.

## 6. Conclusions

Using eleven readily available clinical factors, we developed and validated a risk score that accurately predicts the ten-year risk of cardiovascular events in a cohort of HIV-infected patients. This novel tool creates a promising opportunity to prevent cardiovascular disease, one of the conditions associated with morbidity and mortality among HIV-infected patients living longer during the era of antiviral therapy [1]. Future studies can include validation of our risk model using other HIV-infected cohorts.

## Figures and Tables

**Figure 1 cells-12-00523-f001:**
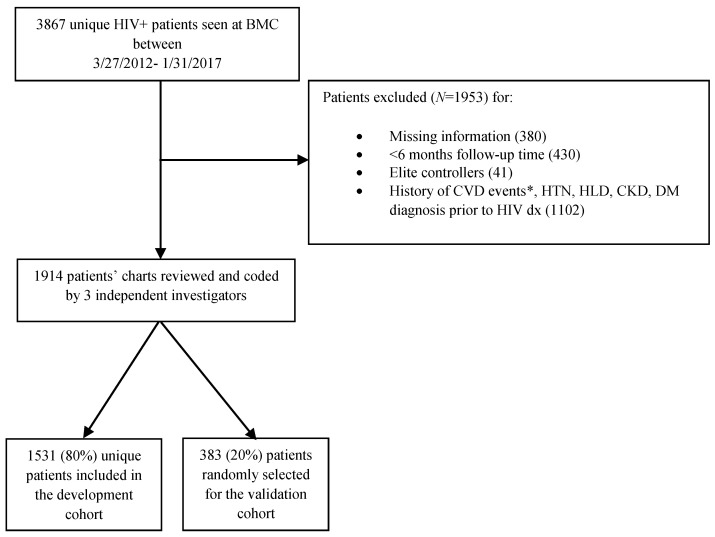
Flow chart showing the selection process used to create development and validation cohorts. BMC: Boston Medical Center; CVD: Cardiovascular disease; HTN: Hypertension; HLD: Hyperlipidemia, CKD: chronic kidney disease; DM: diabetes; HIV: human immunodeficiency virus. * CVD events included sudden cardiac deaths, hospitalizations for unstable angina, myocardial infarctions, strokes, transient ischemic attacks, carotid endarterectomies, and coronary artery bypass graft surgeries.

**Figure 2 cells-12-00523-f002:**
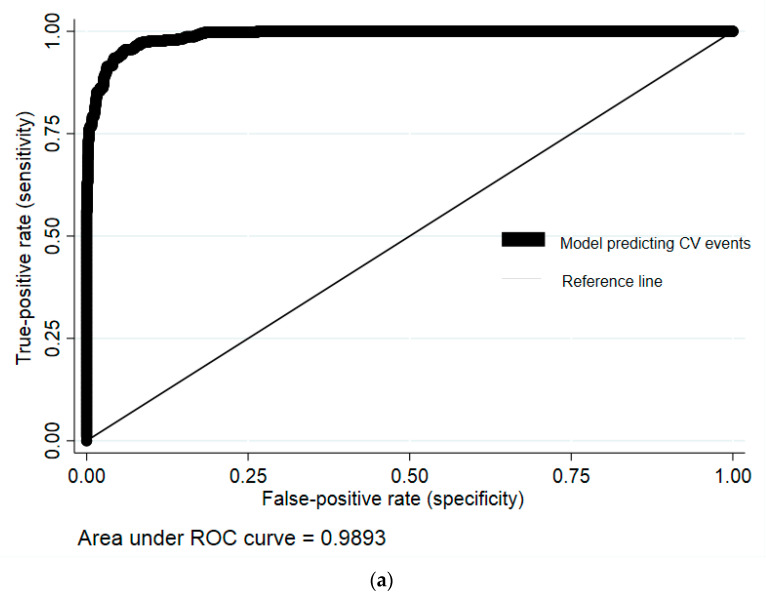
(**a**) Receiver operating characteristics (ROCs) curve for the diagnosis of cardiovascular (CV) events using the development cohort. Area under the ROC curve yielded a value of 0.989 suggesting excellent discrimination between CV events vs. no events. Variables included male sex, African American race, current age, presence of hypertension, hyperlipidemia, chronic kidney disease, diabetes, smoking, age at HIV diagnosis, peak HIV viral load, and nadir CD4 lymphocyte count. (**b**) Receiver operating characteristics (ROCs) curve for the diagnosis of cardiovascular (CV) events using the validation cohort. Area under the ROC curves provides a value of 0.957, indicating excellent discrimination between CVD events vs. no events. Variables included male sex, African American race, current age, hypertension, hyperlipidemia, chronic kidney disease, diabetes, smoking, age at HIV diagnosis, peak HIV viral load, and nadir CD4 lymphocyte count.

**Table 1 cells-12-00523-t001:** Demographic, clinical, and laboratory characteristics of patients included in the development cohort (*n* = 1531).

Patients Characteristics	
**Age, years (mean + /− SD)**	46.1 +/− 11.2
**Age at HIV diagnosis**, years (mean + /− SD)	37.1 +/− 10.0
**Gender**	
Female (%)	654 (42.8)
**Race**	
White (%)	629 (41.1)
African American (%)	649 (42.4)
Hispanic/Latino (%)	201 (13.1)
Asian (%)	52 (3.4)
**Hypertension (%)**	378 (24.7)
**Hyperlipidemia (%)**	364 (23.8)
**Chronic Kidney Disease (%)**	378 (24.7)
**Diabetes Mellitus (%)**	406 (26.5)
**Hepatitis C (%)**	301 (19.7)
**Smoking (%)**	501 (32.7)
**Substance use disorder (%)**	582 (38.0)
**Alcohol use disorder (%)**	367 (24.0)
**Cardiovascular Events**	**No. of Patients (%)**
Sudden cardiac death	51 (3.3)
Hospitalization for unstable angina	21 (1.4)
Myocardial infarction	135 (8.8)
Stroke	51 (3.3)
TIA	55 (3.6)
Carotid endarterectomy	38 (2.5)
CABG	33 (2.2)
**Total CVD events**	**384 (25.1)**
**CD4 count at HIV diagnosis**, cells/μL (mean, SD)	165 (47)
**CD4% at HIV diagnosis** (mean, SD)	12.2 (6.2)
**CD4 nadir**, cells/μL (mean, SD)	114.2 (57.4)
**CD4% nadir** (mean, SD)	9.3 (3.9)
**Peak HIV Viral Load**, copies/mL (mean, SD)	280,149 (1,216,258)
**HIV Viral Load at HIV diagnosis**, copies/mL (mean, SD)	210,147 (678,800)
**Months to control HIV** (mean, SD)	5.7 (5.2)
**Months from HIV diagnosis to CVD event** (mean, SD)	81.4 (46.2)
**No. regimens necessary to reach an undetectable status** (median, range)	1 (1–4)
**Adherence to ARVs** (median, range)	3 (1–4)
**Abacavir treatment for >6 months** (%)	306 (20.1)

ARVs: antiretrovirals; CABG: coronary artery bypass grafting; CD4: cluster of differentiation 4; CVD: cardiovascular diseases; HIV: human immunodeficiency virus; TIA: transient ischemic attack.

**Table 2 cells-12-00523-t002:** Multivariable logistic regression model of statistically significant variables used to predict cardiovascular events in the development cohort (P_model_ ≤ 0.0001).

Variables Associated with Cardiovascular Events	Coefficient	95% CI	*p*-Value
**Sex** (male)	3.26	2.53–3.99	**<0.001**
**Race**			
White	Reference		
African American	2.32	1.50–3.13	**<0.001**
Latino	−0.75	−1.93–0.42	0.207
Asian	−0.62	−3.60–0.73	0.598
**Current age**	0.10	0.07–0.13	**<0.001**
**Age at HIV diagnosis**	−0.06	−0.10–(−0.02)	**0.006**
**Hypertension**	0.87	0.20–1.54	**0.011**
**Diabetes**	1.25	0.61–1.90	**<0.001**
**Smoking**	1.25	0.12–2.39	**0.030**
**Hyperlipidemia**	3.82	3.03–4.60	**<0.001**
**Chronic kidney disease**	1.943	1.26–2.62	**<0.001**
**Peak HIV viral load**	2.00 × 10^−6^	9.89 × 10^−7^–3.00 × 10^−6^	**<0.001**
**Nadir CD4 count**	−0.02	−0.03–(−0.02)	**<0.001**

**Table 3 cells-12-00523-t003:** Scoring system used for each risk factor.

Risk Factor	Categories	Points
**Sex**	Female	0
	Male	3
**Race**	White	0
	African American	2
**Hypertension**	No	0
	Yes	1
**Diabetes**	No	0
	Yes	1
**Hyperlipidemia**	No	0
	Yes	4
**Chronic Kidney Disease**	No	0
	Yes	2
**Smoking**	No	0
	Yes	1
**Peak HIV Viral Load (copies/mL)**	<20,000	0
	20,000–99,999	1
	100,000–199,999	2
	≥200,000	3
**Current age (years)**	18–30	0
	30–39	1
	40–49	2
	50–59	3
	60–69	4
	≥70	5
**Age at HIV diagnosis (years)**	<24	3
	25–44	2
	45–54	1
	≥55	0
**Nadir CD4 count (cells/mm^3^)**	<50	4
	50–119	2
	120–157	1
	≥158	0

CD4: cluster of differentiation 4; HIV: human immunodeficiency virus.

**Table 4 cells-12-00523-t004:** Risk estimate for developing CV events within 10 years per total points for HIV-infected patients.

Total Points	Risk Estimate (%)
0	0.61
1	0.91
2	1.36
3	2.03
4	3.03
5	4.43
6	6.49
7	9.42
8	13.48
9	18.92
10	25.89
11	34.36
12	43.95
13	54.01
14	63.76
15	72.49
16	79.78
17	85.53
18	89.85
19	92.99
20	95.21
21	96.75
22	97.81
23	98.52
24	99.01
25	99.34
26	99.56
27	99.70
28	99.80
29	99.87

## Data Availability

Data is contained within the article or Appendix A.

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
