# Peer review of "Development and Validation of the HIV-CARDIO-PREDICT Score to Estimate the Risk of Cardiovascular Events in HIV-Infected Patients"

_cells, 2023, doi:10.3390/cells12040523_

Round 1

Reviewer 1 Report

It is an interesting study that developed and validated the CVD prediction model in HIV-infected patients.

Several comments to be considered.

1. For Figure 1, I recommend put the number of participants for validation cohort on the right of those for development cohort. That is because they are also participants, and the current vision looks that they were excluded.

2. For Potential predictors, the authors used current age. How about those who were died before Jan 31, 2017? What is the rational of use current age instead of age at HIV diagnosis?

3. For Outcome, were all the CVD outcomes based on hospitalisation records or death certificate? Are there any missing in data linkage?

4. For the Discussion section, I recommend the authors brief summarise their key findings as a shorter paragraph first so that readers could easily catch their results. 

Reviewer 2 Report

The study has an important limitation regarding the motodology.

Besides, there is a lack of validation in an external cohort and that it is a retrospective study.

Secondly, variables included in the score such as peak HIV viral load and nadir CD4, not always are available in the patient record, what could limit the interpretation of the score and even makes it invalid.

Round 2

Reviewer 2 Report

Although,  it is outside my expertise area, I think that the methods section should be improved and a statistician could review the paper too.  

Authors could consider validate the score in an external cohort. 
